# Learning Stable Allocations of Strictly Convex Stochastic Cooperative Games

## Abstract

Reward allocation has been an important topic in economics, engineering, and machine learning. An important concept in reward allocation is the core, which is the set of stable allocations where no agent has the motivation to deviate from the grand coalition. In previous works, computing the core requires the complete knowledge of the game. However, this is unrealistic, as outcome of the game is often partially known and may be subject to uncertainty. In this paper, we consider the core learning problem in stochastic cooperative games, where the reward distribution is unknown. Our goal is to learn the expected core, that is, the set of allocations that are stable in expectation, given an oracle that returns a stochastic reward for an enquired coalition each round. Within the class of strictly convex games, we present an algorithm that returns a point in the expected core given a polynomial number of samples, with high probability. To analyse the algorithm, we develop a new extension of the separation hyperplane theorem for multiple convex sets.

## 1. Introduction

The reward allocation problem is a fundamental challenge in cooperative games that seeks reward allocation schemes to motivate agents to collaborate or satisfy certain constraints, and its solution concepts have recently gained popularity within the machine learning literature through its application in explainable AI [14, 23, 10, 26] and cooperative Multi-Agent Reinforcement Learning [24, 9, 25]. A crucial notion of reward allocation is stability, defined as an allocation scheme wherein no agent has the motivation to deviate from the grand coalition. The set of stable allocations is called *the core of the game*.

In the classical setting, the reward function is deterministic and commonly known among all agents, with no uncertainty within the game. However, assuming perfect knowledge of the game is often unrealistic, as the outcome of the game may contain uncertainty. This led to the study of stochastic cooperative games, dated back to the seminal works of [6, 22], where stability can be satisfied either with high probability, known as the robust core, or in expectation, known as the expected core. However, in these works, the

distribution of stochastic rewards is given, allowing agents to calculate the reward allocations before the game starts, which is not practical since the knowledge of the reward distribution may only be partially known to the players. When the distribution of the stochastic reward is unknown, the task of learning the stochastic core by sequentially interacting with the environment appears much more challenging.

In our work, we focus on learning the expected core, which circumvents the potential emptiness of the robust core in many practical cases. Moreover, where the stochastic rewards of all coalitions are observed each round, we consider the bandit feedback setting, where only the stochastic reward of the inquired coalition is observed each round. Given the lack of knowledge about the probability distribution of the reward function, learning the expected core using data-driven approaches with bandit feedback is challenging.

Against this background, the contribution of this paper is three-fold: **(1)** We focus on expected core learning problem with unknown reward function, and propose a novel algorithm called the `Common-Points-Picking` algorithm, the first of its kind that is designed to learn the expected core with high probability. Notably, this algorithm is capable of returning a point in an unknown simplex, given access to the stochastic positions of the vertices, which can also be used in other domains, such as convex geometry. **(2)** We establish an analysis for finite sample performance of the `Common-Points-Picking` algorithm. The key component of the analysis revolves around a novel extension of the celebrated hyperplane separation theorem, accompanied by further results in convex geometry, which can also be of independent interest. **(3)** We show that our algorithm returns a point in expected core with at least $1 - \delta$ probability, using $\text{poly}(n, \log(\delta^{-1}))$ number of samples.

## 2. Related Work

**Stochastic Cooperative Games.** The study of stochastic cooperative games can be traced back to at least [6, 22, 21]. The main goal of the allocation scheme is to minimise the probability of objections arising after the realisation of the rewards. These seminal works require information about the reward distribution to compute a stable allocation scheme before the game starts. Stochastic cooperative games have also been studied in a Bayesian setting in a series of papers [2, 4, 5, 3], where the distribution of the reward is

conditioned on a hidden parameter following a prior distribution, which is common knowledge among agents. In contrast to previous works, our paper focuses on studying scenarios where the reward distribution or prior knowledge is not disclosed to the principal agent and computing a stable allocation requires a data-driven method.

**Learning the Core.** The literature on learning the core through sample-based methods can be categorised based on the type of core one seeks to evaluate. Two main concepts of the stochastic core are commonly considered, namely the robust core (i.e. core constraints are satisfied with high probability) [8, 18, 15] and the expected core (i.e. core constraints are satisfied in expectation) [7, 16]. In this work, we investigate the learnability of the expected core, which mitigate the potential emptiness of the robust core [7]. The work most closely related to ours is [16], in which the authors introduce an algorithm designed to approximate the expected core using a robust optimization framework. In the context of full information feedback, where rewards for all allocations are revealed each round, the algorithm in [16] demonstrates asymptotic convergence to the expected core. In contrast, we consider bandit feedback, where applying the algorithm of [16] may result in an exponential number of samples in terms of the number of players (see Appendix E.1 for a detailed explanation). Different than general framework in [16], we propose a novel algorithm that explicitly exploits geometric properties of (strictly) convex game to seek a point in expected core with only $\text{poly}(n)$ number of sample, with high probability.

## 3. Problem Description

### 3.1. Preliminaries

**Notations.** For $k \in \mathbb{N}^+$, denote $[k]$ as set $\{1, 2, \dots, k\}$. For $n \in \mathbb{N}^+$, let $\mathbf{E}^n$ be the $n$-dimensional Euclidean space, and let us denote $\mathcal{D}$ as the Euclidean distance in $\mathbf{E}^n$. Denote $\mathbf{1}_n$ as the vector $[1, \dots, 1] \in \mathbb{R}^n$. Denote $\langle \cdot, \cdot \rangle$ as the dot product. For a set $C$, we denote $C \setminus x$ as the set resulting from eliminating an element $x$ in $C$. For $C \subset \mathbf{E}^n$, let $\text{diam}(C) := \max_{x,y \in C} \mathcal{D}(x, y)$, and $\text{Conv}(C)$ denote the diameter and the convex hull of $C$, respectively. Denote $\mathfrak{S}_n := \{\omega : [n] \to [n] \mid \omega \text{ is a bijection}\}$ as the permutation group of $[n]$. For any collection of permutations $\mathcal{P} \subset \mathfrak{S}_n$, we denote $\omega_p, \; p \in [|\mathcal{P}|]$, as $p^{\text{th}}$ permutation order in $\mathcal{P}$. Given a set $C$, we denote by $\mathcal{M}(C)$ the space of all probability distributions on $C$.

**Stochastic Cooperative Games.** A *stochastic* cooperative game is defined as a tuple $(N, \mathbb{P})$, where $N$ is a set containing all agents with number of agents to be $|N| = n$, and $\mathbb{P} = \{\mathbb{P}_S \in \mathcal{M}([0,1]) \mid S \subseteq N\}$ is the collection of reward distributions with $\mathbb{P}_S$ to be the reward distribution w.r.t. the coalition $S$. For any coalition $S \subseteq N$,

we denote $\mu(S) := \mathbb{E}_{r \sim \mathbb{P}_S}[r]$ as the expected reward of coalition $S$. For a reward allocation scheme $x \in \mathbb{R}^n$, let $x(S) := \sum_{i \in S} x_i$ as the total reward allocation for players in $S$. A reward allocation $x$ is *effective* if $x(N) = \mu(N)$. The hyperplane of all effective reward allocations, denoted by $H_N$, is defined as $H_N = \{x \in \mathbb{R}^n \mid x(N) = \mu(N)\}$. The (strictly) convex stochastic cooperative game can be defined as follows:

**Definition 1** ($\varsigma$-**Strictly convex cooperative game**). For some constant $\varsigma \geq 0$, A stochastic cooperative game is convex if the expected reward function is supermodular [19], that is, $\forall i \notin S \cup C$; and $\forall C \subseteq S \subseteq N$,

$$\mu(S \cup \{i\}) - \mu(S) \geq \mu(C \cup \{i\}) - \mu(C) + \varsigma. \quad (1)$$

When $\varsigma = 0$, we simply call the game convex, otherwise, it is strictly convex. Next, we define the expected core as follows:

**Definition 2** (**Expected core** [16]). The core is defined as

$$\text{E-Core} := \{x \in \mathbb{R}^n \mid x(N) = \mu(N);$$
$$x(S) \geq \mu(S), \; \forall S \subseteq N\}.$$

Note that, as $\text{E-Core} \subset H_N$, its dimension is at most $(n - 1)$. We say that E-Core is *full dimensional* whenever its dimension is $n - 1$. For any $\omega \in \mathfrak{S}_n$, define the marginal vector $\phi^\omega \in \mathbb{R}^n$ corresponding to $\omega$, that is, its $i^{\text{th}}$ entry is

$$\phi_i^\omega := \mu(P^\omega(i)) - \mu(P^\omega(i) \setminus i), \quad (2)$$

where $P_i^\omega = \{j \mid \omega(j) \leq \omega(i)\}$. In convex games, each vertex of the core in the convex game is a marginal vector corresponding to a permutation order [19]. This is a special property of convex games, which plays a crucial role in our algorithm design.

### 3.2. Problem Setting

In our setting we assume that there is a principal who does not know the reward distribution $\mathbb{P}$. In each round $t$, the principal queries a coalition $S_t \subset N$. The environment returns a vector $r_t \sim \mathbb{P}_{S_t}$ independently of the past. For simplicity, we assume that the agent knows the expected reward of the grand coalition $\mu(N)$. Our question is how many samples are needed so that with high probability $1 - \delta$, one can compute a point $x \in \text{E-Core}$.

As well shall show in Theorem 5, if E-Core is not full-dimensional, no algorithm can output a point in E-Core with finite samples. As such, to guarantee the learnability of the E-Core. From now on in the rest of this paper, we assume that:

**Assumption 3.** *The game is $\varsigma$-strictly convex.*

Note that *strict* convexity immediately implies full dimensionality [19], which is not the case with convexity.

## 4. `Common-Points-Picking` algorithm

In deterministic convex game, to compute a point in the core, one can query a vertex of the E-Core, that is, a marginal vector corresponding to a permutation order $\omega \in \mathfrak{S}_n$ [19]. Given that the game is now stochastic, this approach is no longer applicable as we can only compute the confidence set instead of the exact position of the vertex. One approach to overcome the effect of uncertainty is to estimate multiple vertices of the E-Core. Let $\mathcal{P} \subset \mathfrak{S}_n$ be a collection of permutations, $Q = \{\phi^{\omega_p} \mid \omega_p \in \mathcal{P}\}$ be the set of vertices corresponding to $\mathcal{P}$, and $\mathcal{C}_p \ni \phi^{\omega_p}$ is the confidence set. It is clear that $\mathrm{Conv}(Q) \subset$ E-Core, since $Q$ is a subset of vertices of E-Core. The challenge is ensuring the algorithm outputs a point within the convex hull of any points in the confidence sets, since the true vertex position can be anywhere within these sets. A sufficient condition to achieve this is that, given $|\mathcal{P}|$ confidence sets $\{\mathcal{C}_p\}_{p \in [|\mathcal{P}|]}$, for each $x^p \in \mathcal{C}_p$,

$$\bigcap_{\substack{x^p \in \mathcal{C}_p \\ p \in [|\mathcal{P}|]}} \mathrm{Conv}\left(\{x^p\}_{p \in [|\mathcal{P}|]}\right) \neq \varnothing. \tag{3}$$

This condition means that there exists a common point among all the convex hulls formed by choosing any point in confidence sets, $x^p \in \mathcal{C}_p$. We call the above intersection a *set of common points*. It is clear that set of common points is a subset of the E-Core. We first state a necessary condition for the number of vertices of E-Core need to estimate for (3) can be satisfied:

**Proposition 4.** *Suppose that all the confidence sets are full dimensional, i.e., $\dim(\mathcal{C}_p) = n - 1$, $\forall p \in [|\mathcal{P}|]$, and suppose that $|\mathcal{P}| < n$. There does not exist common point.*

Proposition 4 implies that one needs to estimate at least $n$ vertices to guarantee the existence of a common point. As such, from now on, we assume that $|\mathcal{P}| = n$. Based on the above intuition, we propose `Common-Points-Picking`, whose pseudo code is described in Algorithm 1, 2.

Before explaining our algorithm, let us construct the confidence sets using Hoeffding's inequality as follows. Let $r_{\mathrm{ep}}(\varnothing) = 0$, $\forall \mathrm{ep} > 0$, define the empirical marginal vector w.r.t. permutation $\omega_p$ as $\hat{\phi}^{\omega_p} \in \mathbb{R}^n$ at epoch ep as

$$\hat{\phi}_i^{\omega_p}(\mathrm{ep}) = \frac{1}{\mathrm{ep}} \left( \sum_{s=1}^{\mathrm{ep}} r_s\left(P_i^{\omega_p}\right) - r_s\left(P_i^{\omega_p} \setminus i\right) \right). \tag{4}$$

By the Hoeffding's inequality, one has that after ep epochs, $\forall \omega_p \in \mathcal{P}$, with probability at least $1 - \delta$, $\phi^{\omega_p}$ lies in

$$\mathcal{C}_p := \left\{ x \in H_N \;\middle|\; \left\| x - \hat{\phi}^{\omega_p} \right\|_\infty \leq b_{\mathrm{ep}} \right\};$$

$$\text{s.t.} \quad b_{\mathrm{ep}} := \sqrt{\frac{2 \log(n \, \mathrm{ep} \, \delta^{-1})}{\mathrm{ep}}}. \tag{5}$$

---

**Algorithm 1** Common Points Picking

1: Input collection of permutation order $\mathcal{P} = \{\omega_p\}_{p \in [n]}$.
2: $t = 0$, ep $= 0$, $\mathcal{C}_p = \varnothing, \forall p \in [n]$.
3: **while** `Stopping-Condition`$\left(\{\mathcal{C}_p\}_{p \in [n]}, b_{\mathrm{ep}}\right)$ **do**
4:    ep $\leftarrow$ ep $+ 1$;
5:    **for** $p \in [n]$ **do**
6:       **for** $i \in [n]$ **do**
7:          Query $P_i^{\omega_p}$.
8:          Orcale returns $r_{\mathrm{ep}}\left(P_i^{\omega_p}\right) \leftarrow r_t$.
9:          $t \leftarrow t + 1$.
10:         Computing $\hat{\phi}_i^{\omega_p}(\mathrm{ep})$ as (4).
11:       **end for**
12:    **end for**
13:    $\forall p \in [n]$, Compute confidence set $\mathcal{C}_p$, $b_{\mathrm{ep}}$ as (5).
14: **end while**
15: Return $x^\star = \frac{1}{n} \sum_{p \in [n]} \hat{\phi}^{\omega_p}(\mathrm{ep})$.

---

**Algorithm 2** Stopping Condition

1: Input collection $\{\mathcal{C}_p\}_{p \in [n]}$, and confidence bonus $b_{\mathrm{ep}}$.
2: Compute $\epsilon_{\mathrm{ep}} = 2\sqrt{n} b_{\mathrm{ep}}$.
3: **if** $\mathcal{C}_p = \varnothing$ for some $p \in [n]$ **then**
4:    Return FALSE.
5: **end if**
6: **for** $p \in [n]$ **do**
7:    Computing separating hyperplane $H_p$ between $\mathcal{C}_p$ and $\{\mathcal{C}_q\}_{q \neq p}$ as eq (7).
8:    Computing distance: $h_p := \mathcal{D}(\mathcal{C}_p, H_p)$.
9:    **if** $h_p < n \, \epsilon_{\mathrm{ep}}$ **then**
10:       Return FALSE.
11:    **end if**
12: **end for**
13: Return TRUE.

---

The `Common-Points-Picking` Algorithm (Algorithm 1) can be described as follows. In each epoch ep, assuming that the stopping condition is not satisfied, the algorithm estimates the marginal vectors corresponding to the collection of given permutation orders $\{\hat{\phi}^{\omega_p}(\mathrm{ep})\}_{p \in [n]}$ (lines 6-10). Next, it calculates the confidence bonus $b_{\mathrm{ep}}$, the confidence sets $\{\mathcal{C}_p\}_{p \in [n]}$, and checks the stopping condition for the next epoch. The algorithm continues until the stopping condition is satisfied, and then returns the average of the most recent values of the marginal vectors in $\mathcal{P}$.

The termination of the `Common-Points-Picking` algorithm is based on the `Stopping-Condition` algorithm (Algorithm 2), which can be described as follows. For each confidence set $\mathcal{C}_p$, the algorithm attempts to calculate the separating hyperplane $H_p$, that separates $\mathcal{C}_p$ from the rest $\{\mathcal{C}_q\}_{q \neq p}$ (line 7). After computing $H_p$, the algorithm checks whether the distance from the confidence set $\mathcal{C}_p$ to $H_p$ is large enough (lines 8, 9). It checks for all $p \in [n]$; if no condition is violated, then the algorithm returns TRUE.

## 5. Main Results

Before proceeding to the analysis of Algorithm 1, let us exclude the case where learning a stable allocation is not possible, thereby emphasizing the need of the *strict* convexity assumption.

**Theorem 5.** *Suppose that E-Core has dimension $k < n - 1$, for any $0.2 > \delta > 0$ and with finite samples, no algorithm can output a point in E-Core with probability at least $1 - \delta$.*

We note that convex games may have a low-dimensional core (e.g., Example 13 in Appendix A). This suggests that convexity alone does not ensure the problem's learnability, emphasizing the requirement for strict convexity.

### 5.1. On the Stopping Condition

In this subsection, we explain the construction of the stopping condition in Algorithm 2. To simplify the presentation, we restrict our attention to $H_N$ and consider it as $\mathbf{E}^{n-1}$. First, we state a necessary condition for the existence of common points.

**Proposition 6.** *Suppose there is a $(n-2)$-dimensional hyperplane that intersect with all the interior of confidence sets $\mathcal{C}_p$, $\forall p \in [n]$, then common points do not exist.*

Proposition 6 suggests that if the ground truth simplex $\mathrm{Conv}(Q)$ is not full-dimensional, then the common set is empty. On the other hand, when the confidence sets are well-arranged and sufficiently small, that is, there does not exist a hyperplane that intersects with all of them, a nice separating property emerges, as stated in the next theorem. This *new result can be considered as an extension of the classic separating hyperplane theorem* [1].

**Theorem 7** (**Hyperplane separation theorem for multiple convex sets**). *Assume that $\{\mathcal{C}_p\}_{p\in[n]}$ are mutually disjoint compact and convex subsets in $\mathbf{E}^{n-1}$. Suppose that there does not exist a $(n-2)$-dimensional hyperplane that intersects with confidence sets $\mathcal{C}_p$, $\forall p \in [n]$, then for each $p \in [n]$, there exists a hyperplane that separates $\mathcal{C}_p$ from $\bigcup_{q \neq p} \mathcal{C}_q$.*

When those confidence sets are well-separated, we can provide a sufficient condition for that the common points exist. Let $H_p$ be a separating hyperplane that separate $\mathcal{C}_p$ from $\bigcup_{q \neq p} \mathcal{C}_q$. We define $H_p$ corresponding with tuple $(v^p, c^p)$, where $v^p$ is a unit normal vector of $H_p$ and $c^p$ is a scalar. Now, denote $E_p = \{x \in \mathbf{E}^{n-1} \mid \langle v^p, x \rangle < c^p\}$ as the half space containing $\mathcal{C}_p$. We have that:

**Lemma 8.** *For any $x^p \in \mathcal{C}_p$, $p \in [n]$,*

$$\bigcap_{p\in[n]} E_p \subseteq \mathrm{Conv}\left(\{x^p\}_{p\in[n]}\right). \tag{6}$$

*Consequently, if $\bigcap_{p\in[n]} E_p$ is nonempty, it is the subset of common points.*

The key implication here is that Lemma 8 provides us a method to find a point in the common set. Using Lemma 8, we can show that if each distance from a confidence set to its separating hyperplane is sufficiently large compared to the diameter of the other confidence sets, then a common point exists, as stated in the following theorem.

**Theorem 9.** *Given a collection of confident set $\{\mathcal{C}_p\}_{p\in[n]}$ and let $Q = \{x^p\}_{p\in[n]}$, for any $x^p \in \mathcal{C}_p$. For any $p \in [n]$, denote $H_p(Q)$ as the $(n-1)$-dimensional hyperplane with constant $(v^p, c^p)$, $\|v^p\| = 1$ such that*

$$\begin{cases} \langle v^p, x \rangle = c^p + \max_{q\in[n]\backslash p} \mathrm{diam}(\mathcal{C}_p), & \forall x \in Q \setminus x^p. \\ \langle v^p, x^p \rangle < c^p + \max_{q\in[n]\backslash p} \mathrm{diam}(\mathcal{C}_p). \end{cases} \tag{7}$$

*For all $p \in [n]$, if the following holds*

$$\mathcal{D}(\mathcal{C}_p, H_p(Q)) > 2n \left( \max_{q\in[n]\backslash p} \mathrm{diam}(\mathcal{C}_q) \right); \tag{8}$$

*then, $x^\star = \frac{1}{n} \sum_{p\in[n]} x^p$ is a common point.*

### 5.2. Sample Complexity Analysis

In strictly convex game, we show that the conditions of Theorem 9 can be satisfied with high probability (see Appendix C). This upper-bounds the sample complexity as follows.

**Theorem 10.** *Suppose that Assumption 3 holds. There exists a choice of collection of permutation order $\mathcal{P}$, such that for any $\delta \in [0, 1]$, if the number of samples is bounded by*

$$T = O\left(\frac{n^{15} \log(n\delta^{-1}\varsigma^{-1})}{\varsigma^4}\right), \tag{9}$$

*then* `Common-Points-Picking` *algorithm returns a point in E-Core with probability at least $1 - \delta$.*

We describe the choice of $\mathcal{P}$ in Appendix C, along with several different choice of collection of $n$ vertices that probably achieve better scaling with $n$ for large class of the game.

## 6. Conclusion and Future Work

In this paper, we address the challenge of learning the expected core of a strictly convex stochastic cooperative game. Under the assumptions of strict convexity and a large interior of the core, we introduce an algorithm named `Common-Points-Picking` to learn the expected core. Our algorithm guarantees termination after $\mathrm{poly}\left(n, \log(\delta^{-1}), \varsigma^{-1}\right)$ samples and returns a point in the expected core with probability $(1 - \delta)$. For future work, we will investigate whether the sample complexity of our algorithm can be further improved by incorporating adaptive sampling techniques into the algorithm.

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

# Contents of Appendix

## A. Proof of Theorem 5

Here and onwards, we adopt the following notation convention: for real numbers $a, b \in [0, 1]$, $\mathrm{KL}(a, b)$ represents the KL-divergence $\mathrm{KL}(p, q)$ where $p, q$ are probability distributions on $\{0, 1\}$ such that $p(1) = a$, $q(1) = b$. In other words,

$$\mathrm{KL}(a, b) = a \ln\left(\tfrac{a}{b}\right) + (1 - a) \ln\left(\tfrac{1-a}{1-b}\right).$$

**Lemma 11** ([13]). *For any $0 < \varepsilon < y \le 1$, $\mathrm{KL}(y - \varepsilon, y) < \varepsilon^2 / y(1 - y)$.*

Before stating the proof of Theorem 5, let us introduce some extra notations. Given a game $G = (N, \mathbb{P})$, with the expected reward function $\mu$, we define the following.

- $H_C(G) := \{x \in \mathbb{R}^n \mid x(C) = \mu(C)\}$ is the hyperplane corresponding to the effective allocation w.r.t coalition $C$.

- $\text{E-Core}(G)$ is the expected core of the game $G$.

- $F_C(G) := \text{E-Core}(G) \cap H_{N\setminus C}(G)$ is facet of the E-Core corresponding to the coalition $C$.

We use the following definition of the face games in Theorem 5, introduced by [11].

**Definition 12** (Face Game). Given a game $G = (N, \mathbb{P})$ with $\mu(S) = \mathbb{E}_{r \sim \mathbb{P}_S}[r]$, $\forall S \subset N$. For any $C \subset N$, define a face game $G(C) = (N, \mathbb{P}^C)$ with $\mu_{F_C}(S) = \mathbb{E}_{r \sim \mathbb{P}_S^C}[r]$ such that, for any $S \subset N$,

$$\mu_{F_C}(S) = \mu((S \cap C) \cup (N \setminus C)) - \mu(N \setminus C) + \mu(S \cap (N \setminus C)). \tag{10}$$

[11] showed that the expected core of $G(C)$ is exactly the facet of E-Core$(G)$ corresponding $C$, that is, E-Core$(G(C)) = F_C(G)$. As noted in [27], one can decompose the reward function of the face game as follows. For any $S \subset N$, we have that

$$\mu_{F_C}(S) = \mu_{F_C}(S \cap C) + \mu_{F_C}(S \cap (N \setminus C)). \tag{11}$$

We now proceed the proof of Theorem 5.

**Proof of Theorem 5.** Denote the set convex games with Bernolli reward as **GB**, that is,

$$\mathbf{GB} = \{G = (N, \mathbb{P}) \mid \mathbb{P} = \{\mathbb{P}_S\}_{S \subseteq N}; \ \mathbb{P}_S \in \mathcal{M}(\{0,1\}), \ \forall S \subseteq N\}.$$

**Face-game instances and the distance between their E-Core.** We first define two games, $G_0$ and $G_1$, with a full-dimensional E-Core, such that $G_1$ is a slight perturbation of $G_0$. Next, we define face games corresponding to $G_0$ and $G_1$ using the perturbed facet. We then show that the distance between the cores of these two face games is at least some positive number $\varepsilon > 0$.

Define a strictly convex game $G_0 := (N, \mathbb{P}) \in \mathbf{GB}$, such that $\mu^0(S) := \mathbb{E}_{r \sim \mathbb{P}_S}[r]$, and assume that $\mu^0$ is $\varsigma$-strictly supermodular. Now, fix a subset $C \subset N$, let define a perturbed game instance $G_1 := (N, \mathbb{Q}) \in \mathbf{GB}$, with $\mu^1(S) := \mathbb{E}_{r \sim \mathbb{Q}_S}[r]$ such that

$$\begin{cases} \mu^1(C) := \mu^0(C) - \varepsilon; \\ \mu^1(S) := \mu^0(S); & \forall S \subset N, \ S \neq C; \end{cases} \tag{12}$$

for some $0 < \varepsilon < \varsigma$. It is straightforward that $G_1$ is $(\varsigma - \varepsilon)$-strictly convex. Therefore, E-Core$(G_0)$ and E-Core$(G_1)$ are both full-dimensional.

Fixing a coalition $C \subset N$, we now construct the face games from $G_0, \ G_1$ as in Definition 12.
Let $G_0(C) := (N, \mathbb{P}^C), \ G_1(C) := (N, \mathbb{Q}^C) \in \mathbf{GB}$, whose expected rewards $\mu^0_{F_C}$ and $\mu^1_{F_C}$ are defined by applying (10) to $\mu^0$ and $\mu^1$ respectively. Now, we consider the difference between the expected reward function of these two games.

$$\begin{cases} |\mu^1_{F_C}(S) - \mu^0_{F_C}(S)| = 0 & \forall S \subset N \setminus C \\ |\mu^1_{F_C}(S) - \mu^0_{F_C}(S)| = \varepsilon & \forall S \subseteq C \\ |\mu^1_{F_C}(N \setminus C) - \mu^0_{F_C}(N \setminus C)| = \varepsilon. \end{cases} \tag{13}$$

As one can always decompose the set $S = (S \cap C) \cup (S \cap N \setminus C)$, by the decomposibility of the face game (11), we has that

$$|\mu^1_{F_C}(S) - \mu^0_{F_C}(S)| \leq \varepsilon, \ \forall S \subset N. \tag{14}$$

As the core of face game $G_0(C)$ and $G_1(C)$ lie on the hyperplane corresponding to the coalition $N \setminus C$, and the distance between the hyperplanes of $G_0$ and $G_1$ is $\varepsilon$, which lower bounds the distance between the expected core of $G_0(C)$ and $G_1(C)$. In particular, as E-Core$(G_0(C)) = F_C(G_0)$ and E-Core$(G_1(C)) = F_C(G_1)$, and $|\mu^1(N \setminus C) - \mu^0(N \setminus C)| = \varepsilon$, which leads to $\mathcal{D}(H_{N \setminus C}(G_0), H_{N \setminus C}(G_1)) = \varepsilon$, we have that

$$\mathcal{D}\left(\text{E-Core}(G_0(C)), \text{E-Core}(G_1(C))\right) \geq \varepsilon. \tag{15}$$

**The KL distance and imposibility of learning low-dimensional E-Core.** We show that, with probability $\delta \in (0, 0.2)$, any learner cannot distinguish between $G_0(C)$ and $G_1(C)$ given there are finite number of samples. We use the information-theoretic framework similar which is well developed within multi-armed bandit literature.

We first upper bound the KL-distance between $\mathbb{P}^C_S, \mathbb{Q}^C_S, \ \forall S \subset N$. Denote $c_1 := \min_{S \subset N} \left(\mu^0_{F_C}(S)(1 - \mu^0_{F_C}(S))\right) > 0$, by Lemma 11, we have that

$$\text{KL}\left(\mathbb{P}^C_S, \mathbb{Q}^C_S\right) = \text{KL}\left(\mu^0_{F_C}(S), \mu^1_{F_C}(S)\right) \leq \frac{\varepsilon^2}{c_1}, \quad \forall S \subset N.$$

Define the probability space $\Psi = 2^N \times \{0, 1\}$. Fix any algorithm (possibly randomised) $\mathcal{A}$. At round $t$, denote $(S_t, r_t) \in \Psi$ as the coalition selected by the algorithm and the reward return by the environment. At round $s < t$, denote $\nu^t_0, \ \nu^t_1$ as the probability distribution over $\Psi^t$ determined by $\mathcal{A}$ and $\mathbb{P}, \mathbb{Q}$ accordingly.

We have the following, as stated in the appendix of [13]. For any $u < t$, one has that,

$$
\mathrm{KL}\left(\nu_0^u, \nu_1^u\right) = \sum_{\psi^{u-1} \in \Psi^{u-1}} \nu_0^u(\psi^u) \log \left( \frac{\nu_0^u(\psi^u \mid \psi^{u-1})}{\nu_1^u(\psi^u \mid \psi^{u-1})} \right)
$$

$$
= \sum_{\psi^{u-1} \in \Psi^{u-1}} \nu_0^u(\psi^u) \log \left( \frac{\nu_0^u(S_u \mid \psi^{u-1})}{\nu_1^u(S_u \mid \psi^{u-1})} \cdot \frac{\nu_0^u(r_u \mid S_u, \psi^{u-1})}{\nu_1^u(r_u \mid S_u, \psi^{u-1})} \right)
$$

$$
= \sum_{\psi^{u-1} \in \Psi^{u-1}} \nu_0^u(\psi^u) \log \left( \frac{\nu_0^u(r_u \mid S_u, \psi^{u-1})}{\nu_1^u(r_u \mid S_u, \psi^{u-1})} \right)
$$

[As the distribution of $S_u$ depends only on $\mathcal{A}$, not on the distribution $\nu_0^t$, $\nu_1^t$.]

$$
= \sum_{\psi^{u-1} \in \Psi^{u-1}} \sum_{S_u \in 2^N} \sum_{r_u \in \{0,1\}} \nu_0^u(r_u \mid S_u, \psi^{u-1}) \log \left( \frac{\nu_0^u(r_u \mid S_u, \psi^{u-1})}{\nu_1^u(r_u \mid S_u, \psi^{u-1})} \right) \nu_0^u(S_u, \psi^{u-1})
$$

$$
= \sum_{\psi^{u-1} \in \Psi^{u-1}} \sum_{S_u \in 2^N} \mathrm{KL}\left(\mu_{F_C}^0(S_u), \mu_{F_C}^1(S_u)\right) \nu_0^u(S_u, \psi^{u-1})
$$

$$
\leq \frac{\varepsilon^2}{c_1}.
$$

The last inequality hold because $\mathrm{KL}\left(\mu_{F_C}^0(S), \mu_{F_C}^1(S)\right) \leq \frac{\varepsilon^2}{c_1}$, $\forall S \in 2^N$.

We have that

$$
\mathrm{KL}\left(\nu_0^t, \nu_1^t\right) = \sum_{u=1}^t \mathrm{KL}\left(\nu_0^u, \nu_1^u\right) \leq \frac{t\varepsilon^2}{c_1}. \tag{16}
$$

As we can choose $\varepsilon$ to be arbitrarily small, we can choose $\varepsilon$ such that $\mathrm{KL}\left(\nu_0^t, \nu_1^t\right) \leq 0.1$.

Now, define the event $\mathcal{E}$ as the event that $\mathcal{A}$ outputs a point in E-Core$(G_0(C))$, assume that $\nu_0^t(\mathcal{E})$ with probability at least $0.8$. Note that, as E-Core$(G_0(C)) \cap$ E-Core$(G_1(C)) = \varnothing$, $\mathcal{E}$ represents the event where the algorithm fails to output a stable allocation with the game instance $G_1(C)$. We have that from [13]'s Lemma A.5,

$$
\nu_1^t(\mathcal{E}) \geq \nu_0^t(\mathcal{E}) \exp\left( -\frac{\mathrm{KL}\left(\nu_0^t, \nu_1^t\right) + 1/e}{\nu_0^t(\mathcal{E})} \right) > 0.8 \exp\left( -\frac{0.1 + 1/e}{0.8} \right) > 0.3. \tag{17}
$$

As it holds for any $t > 0$, this means that for any finite number of samples, with probability at least $0.1$, the algorithm will output the incorrect point. $\qquad\square$

It is worth noting that convex games may have a low-dimensional core, as demonstrated in the following example.

**Example 13.** Let $\mu(S) = |S|$ for all $S \subseteq N$. It is easy to verify that $\mu$ is indeed convex. The marginal contribution of any player $i$ to any set $S \subseteq N$ is

$$
\mu(S \cup i) - \mu(S) = 1, \quad \forall S \subset N. \tag{18}
$$

Therefore, the only stable allocation is $\mathbf{1}_n$, which coincides with the Shapley value. Hence, the core is one-point set. According to Theorem 5, since the core has a dimension of $0$ in this case, it is impossible to learn a stable allocation with a finite number of samples.

Example 13 suggests that convexity alone does not ensure the problem's learnability, emphasizing the requirement for strict convexity.

## B. `Common-points-picking` algorithm and the stopping condition

### B.1. On the Necessary Conditions for the Existence of Common Points

***Proof of Proposition 4.*** For each $\mathcal{C}_p$, choose a point in its interior, denote as $x^p$. As there are at most $n - 1$ points $\{x^p\}_{p \in [|\mathcal{P}|]}$, there exists a $(n - 2)$-dimensional hyperplane $H$ that contains $\{x^p\}_{p \in [|\mathcal{P}|]}$. Let $\tilde{H}$ be a hyperplane parallel to $H$ and let the distance $\mathcal{D}(H, \tilde{H})$ be arbitrary small.

As confidence sets are full-dimensional $(n - 1)$, $\tilde{H}$ must also intersect with the interiors of all confidence sets. Since $H$ and $\tilde{H}$ are parallel, any convex hull of points within $H$ and $\tilde{H}$ cannot intersect. Therefore, there is no common point. $\square$

***Proof of Proposition 6.*** The proof spirit is similar to that of Proposition 4.

Let $H$ be the $(n - 2)$-dimensional hyperplane that intersects with the interiors of all confidence sets. Let $\tilde{H}$ be a hyperplane parallel to $H$ and let the distance $\mathcal{D}(H, \tilde{H})$ be arbitrary small.

As confidence sets are full-dimensional, $\tilde{H}$ must also intersect with the interiors of all confidence sets. Since $H$ and $\tilde{H}$ are parallel, any convex hull of points within $H$ and $\tilde{H}$ cannot intersect. Therefore, there is no common point. $\square$

### B.2. Extension of Separation Hyperplane Theorem

First, let us recap the notion of separation as follows.

**Definition 14** (**Separating hyperplane**). Let $C$ and $D$ be two compact and convex subsets of $\mathbf{E}^{n-1}$. Let $H$ be a hyperplane defined by the tuple $(v, c)$, where $v$ is a unit normal vector and $c$ is a real number, such that $\langle x, v \rangle = c$, $\forall x \in H$. We say $H$ separates $C$ and $D$ if $\langle x, v \rangle > c$, $\forall x \in C$; and $\langle y, v \rangle < c$, $\forall y \in D$.

Before stating the proof of Theorem 7, let us discuss its non-triviality.

**Remark 15** (**Non-triviality of Theorem 7**). At a first glance, Theorem 7 may appear as a trivial extension of the classic hyperplane separation theorem due to the following reasoning: Consider the union of all hyperplanes that intersect $\bigcup_{q \neq p} \mathcal{C}_q$, which trivially contains $\bigcup_{q \neq p} \mathcal{C}_q$. Then, by assuming that these hyperplanes do not intersect $\mathcal{C}_p$, the separation between $\mathcal{C}_p$ and $\bigcup_{q \neq p} \mathcal{C}_q$ appears to follow from the classic separation hyperplane theorem. However, there is a flaw in the above reasoning: The union of these hyperplanes is *not necessarily convex*. Therefore, the classic separation hyperplane theorem cannot be applied directly. Instead, employing Carathéodory's theorem, we prove in Theorem 7 by contra-position that if the intersection between $\mathcal{C}_p$ and $\mathrm{Conv}(\bigcup_{q \neq p} \mathcal{C}_q)$ is non-empty, then we can construct a low-dimensional hyperplane that intersects with all the set.

The proof of Theorem 7 is a combination of the classic hyperplane separation theorem and the following lemma.

**Lemma 16.** *Let $\{\mathcal{C}_p\}_{p \in [n]}$ be mutually disjoint compact and convex subsets in $\mathbf{E}^{n-1}$. Suppose there does not exist a $(n - 2)$-dimensional hyperplane that intersects with all confidence sets $\mathcal{C}_p$, $\forall p \in [n]$, then for each $p \in [n]$*

$$\mathcal{C}_p \cap \mathrm{Conv}\left(\bigcup_{q \neq p} \mathcal{C}_q\right) = \varnothing. \tag{19}$$

*Proof.* We prove this lemma by contra-position, that is, if there is $\mathcal{C}_p$ such that

$$\mathcal{C}_p \cap \mathrm{Conv}\left(\bigcup_{p \neq q} \mathcal{C}_q\right) \neq \varnothing;$$

then there exist a hyperplane that intersects with all the $\mathcal{C}_p$, $\forall p \in [n]$.

**First**, assume there is a point $x = \mathcal{C}_p \cap \mathrm{Conv}\left(\bigcup_{q \neq p} \mathcal{C}_p\right)$. By Carathéodory's theorem, there are at most $n$ points $x^k \in \bigcup_{q \neq p} \mathcal{C}_q$ such that

$$x = \sum_{k \in [n]} \alpha_k x^k. \tag{20}$$

As each $x^k \in \mathcal{C}_q$ for some $\mathcal{C}_q$, one can rewrite the equation above as

$$x = \sum_{q \neq p} \sum_{k: \, x^k \in \mathcal{C}_q} \alpha_k x^k. \tag{21}$$

Furthermore, we can write

$$\sum_{x^k \in \mathcal{C}_q} \alpha_k x^k = \tilde{\alpha}_q \tilde{x}^q, \quad \text{in which,} \quad \tilde{x}^q := \frac{\sum_{k: \, x^k \in \mathcal{C}_q} \alpha_k x^k}{\sum_{k: \, x^k \in \mathcal{C}_q} \alpha_k}, \quad \text{and} \quad \tilde{\alpha}_q := \sum_{k: \, x^k \in \mathcal{C}_q} \alpha_k. \tag{22}$$

Since $\mathcal{C}_q$ is convex, $\tilde{x}^q \in \mathcal{C}_q$. Substituting (22) into (20), one obtains

$$x = \sum_{q \neq p} \tilde{\alpha}_q \tilde{x}^q. \tag{23}$$

Define $H$ as a hyperplane that passes through all $\tilde{x}_q$, we have that $x \in H$.

**Second**, we now show how to construct a hyperplane that intersects with all $\mathcal{C}_m$, $m \in [n]$. Let $I$ be the set of indices such that $\mathcal{C}_q \ni \tilde{x}_q$. We have two following cases.

(i) First, if $|I| = n - 1$, then $H$ is the $(n-2)$-dimensional hyperplane that intersect with all $\mathcal{C}_m$, $m \in [n]$.

(ii) Second, if $|I| < n - 1$, for any $\mathcal{C}_{q'} \neq \mathcal{C}_p$ that does not contain any $\tilde{x}^q$, we choose any arbitrary point $x^{q'} \in \mathcal{C}_{q'}$. As there are $n - 1$ points of $\tilde{x}^q$ and $x^{q'}$, there exists a hyperplane $\overline{H}$ that contains all these points. Furthermore, $\overline{H}$ must contain $x$, so it is the $(n-2)$-dimensional hyperplane that intersects with all sets $\mathcal{C}_m$, $\forall m \in [n]$.

$\square$

Now, we state the proof of Theorem 7.

***Proof of Theorem 7.*** As a result of Lemma 16, we have that for all $\mathcal{C}_p, \forall p \in [n]$,

$$\mathcal{C}_p \cap \text{Conv} \left( \bigcup_{q \neq p} \mathcal{C}_q \right) = \varnothing. \tag{24}$$

Therefore, by the hyperplane separation theorem, there must exist a hyperplane that separates $\mathcal{C}_p$ and $\text{Conv} \left( \bigcup_{q \neq p} \mathcal{C}_q \right)$. $\square$

### B.3. Correctness of the Stopping Condition

***Proof of Lemma 8.*** Let us denote $\Delta_n$ as $\text{Conv} \left( \{x^p\}_{p \in [n]} \right)$. As there is no hyperplane of dimension $n - 2$ go through all the set $\mathcal{C}_p$, the simplex $\Delta_n$ is $(n-1)$ dimensional. We have that

$$\bigcap_{p \in [n]} E_p \subseteq \Delta_n \iff \Delta_n^c \subseteq \bigcup_{p \in [n]} E_p^c;$$

where $E_p^c$ is the complement of the set $E_p$.

We will prove the RHS of the above. Consider $\hat{x} \in \Delta_n^c$, as $\Delta_n$ is full dimensional, $\hat{x}$ can be uniquely written as affine combination of the vertices, that is,

$$\hat{x} = \sum_{p \in [n]} \lambda_p x^p, \quad \sum_{p \in [n]} \lambda_p = 1.$$

As $\hat{x} \in \Delta_n^c$, there must exist some $\lambda_k < 0$.

Now, we shall prove $\hat{x} \in E_k^c$. Consider the following,

$$
\begin{aligned}
\langle v^k, \hat{x} \rangle = \left\langle v^k, \sum_{p \in [n]} \lambda_p x^p \right\rangle &= \lambda_k \langle v^k, x^k \rangle + \sum_{p \neq k} \lambda_p \langle v^k, x^p \rangle \\
&> \lambda_k c^k + c^k \sum_{p \neq k} \lambda_p \\
&= c^k
\end{aligned}
\tag{25}
$$

The above inequality holds since $\langle v^k, x^k \rangle < c_k$ and $\lambda_k < 0$. Therefore, $\hat{x} \in E_k^c$. This means that

$$
\Delta_n^c \subseteq \bigcup_{k \in [n]} E_k^c.
\tag{26}
$$

$\square$

***Proof of Theorem 9***. Before proceeding the main proof, we show two simple consequences of the construction of $H_p(Q)$, $p \in [n]$ and the assumption (8).

Fact 1: *Consider $p \in [n]$, $H_p(Q)$ acts as a separating hyperplane for $C_p$.* To see this, assume that $H_p(Q)$ is not a separate hyperplane for $C_p$, then there exists $z^p \in C_p$ such that $\langle v^p, z^p \rangle \geq c^p$. From (7), we have $\langle v^p, x^p \rangle \leq c^p + \max_{q \in [n] \setminus p} \text{diam}(C_p)$. Then, there are two cases. First, assume that $\langle v^p, x^p \rangle \leq c^p$. As $x^p$, $z^p \in C_p$ and $\langle v^p, z^p \rangle \geq c^p$, there must exist a point $x$ in the line segment $[x^p, z^p]$ such that $\langle v^p, x \rangle = c^p$. This means that $\mathcal{D}(C_p, H_p) = 0$, which violates assumption (8). Second, assume that $c^p \leq \langle v^p, x^p \rangle \leq c^p + \max_{q \in [n] \setminus p} \text{diam}(C_p)$. Then, we have that

$$
\mathcal{D}(C_p, H_p) \leq \mathcal{D}(x^p, H_p) = |\langle v^p, x^p \rangle - c^p| \leq \max_{q \in [n] \setminus p} \text{diam}(C_q).
$$

This also violates assumption (8). This implies that if (8) is satisfied, $H_p(Q)$ must separate $C_p$ from $\cup_{q \neq p} C_q$.

Fact 2: *The distance from any point in $C_q$ from $H_p(Q)$ is bounded* as follows. For $x \in C_q$, $q \neq p$, we have that

$$
\mathcal{D}(x, H_p(Q)) \leq \mathcal{D}(x, x^q) + \mathcal{D}(x^q, H_p(Q)) \leq 2 \max_{q' \in [n] \setminus p} \text{diam}(C_{q'}).
\tag{27}
$$

Now, we proceed to the main proof. For the ease of notation, we simply write $H_p$ for $H_p(Q)$.

**First**, from assumption (8), we has that for any $p \in [n]$,

$$
\mathcal{D}(C_p, H_p) = \min_{x \in C_p} \mathcal{D}(x, H_p) = \min_{x \in C_p} |c^p - \langle v^p, x \rangle|.
\tag{28}
$$

We have that

$$
\begin{aligned}
\min_{x \in C_p} \mathcal{D}(x, H_p) &> 2n \max_{q \neq p} \text{diam}(C_q) \\
&\geq \sum_{q \in [n] \setminus p} \max_{x \in C_q} \mathcal{D}(x, H_p).
\end{aligned}
\tag{29}
$$

**Second**, we shows that how to pick a common point which exists when (29) is satisfied. Let us choose a collection of points $x^p \in C_p$, $p \in [n]$, and define

$$
x^\star = \frac{1}{n} \sum_{p \in [n]} x^p.
$$

Now, we show that $x^\star \in E_p$, $\forall p \in [n]$.

For each $p \in [n]$, consider $H_p$. We denote

$$\zeta_{pp} := c^p - \langle v^p, x^p \rangle > 0;$$
$$\zeta_{pq} := \langle v^p, x^q \rangle - c^p > 0, \quad q \neq p.$$

Note that $\mathcal{D}(x, H_p) = |\langle v^p, x \rangle - c^p|$. Follows (29), we have that

$$\zeta_{pp} \geq \min_{x \in \mathcal{C}_p} \mathcal{D}(x, H_p) > \sum_{q \in [n] \backslash p} \max_{x \in \mathcal{C}_q} \mathcal{D}(x, H_p) \geq \sum_{q \in [n] \backslash p} \zeta_{pq}. \tag{30}$$

Now, let consider

$$\langle v^p, x^\star \rangle = \frac{1}{n} \sum_{q \in [n]} \langle v^p, x^q \rangle = \frac{1}{n} \sum_{q \in [n] \backslash p} (c^p + \zeta_{pq}) + \frac{1}{n}(c^p - \zeta_{pp})$$
$$= c^p + \frac{1}{n} \left( \sum_{q \in [n] \backslash p} \zeta_{pq} - \zeta_{pp} \right) < c^p. \tag{31}$$

Therefore, $x^\star \in E_p$. As it is true for all $E_p$, one has that

$$x^\star \in \bigcap_{p \in [n]} E_p. \tag{32}$$

Finally, by Lemma 8, we can conclude that $x^\star$ is a common point. □

Intuitively, Theorem 9 states that if the distance from a confidence set $\mathcal{C}_p$ to the hyperplane $H_p(Q)$ is relatively large compared to the sum of the diameters of all other confidence sets, then the average of any collection of points in the confidence set must be a common point. As such, Theorem 9 determines the stopping condition for Algorithm 1 and provide us a explicit way to find a common point, which validates the correctness of Algorithm 1. In particular, Algorithm 2 checks if conditions (8) are satisfied for the confidence sets in each round. If the conditions are satisfied, then Algorithm 1 stops sampling and returns $x^\star$ as the common point.

## C. On sample complexity of `Common-points-picking` algorithm

Note that while the diameters of confidence sets can be controlled by the number of samples regarding the marginal vector, $\mathcal{D}(\mathcal{C}_p, H_p(Q))$ is a random variable and needs to be handled with care. We show that there exist choices of $n$ vertices such that the simplex formed by them has a sufficiently large width, resulting in the stopping condition being satisfied with high probability after $\text{poly}(n, \varsigma^{-1})$ number of samples.

Now, we show that, the conditions of Theorem 9 can be satisfied with high probability. The distance $\mathcal{D}(\mathcal{C}_p, H_p(Q))$, $p \in [n]$ can be lower bounded by the width of the ground-truth simplex, which is defined as follows:

**Definition 17** (**Width of simplex**). Given $n$ points $\{x^1, ...x^n\}$ in $\mathbb{R}^n$, let matrix $P = [x^i]_{i \in [n]}$, we define the matrix of coordinates of the points in $P$ w.r.t. $x^i$ as $\text{coM}(P, i) := [(x^j - x^i)]_{j \neq i} \in \mathbb{R}^{n \times (n-1)}$. Denote $\sigma_k(M)$ as the $k^{\text{th}}$ singular value of matrix $M$ (with descending order). We define the *width* of the simplex whose coordinate matrix is $P$ as follows

$$\vartheta(P) := \min_{i \in [n]} \sigma_{n-1}(\text{coM}(P, i)). \tag{33}$$

Equipped with the definition of the width, we can bound the distance $\mathcal{D}(\mathcal{C}_p, H_p(Q)), ; p \in [n]$, accordingly as the following lemma.

**Lemma 18.** *Given $n$ points $\{x^1, ..., x^n\}$ in $\mathbb{R}^n$, let $M$ be the matrix corresponding to these points, assume that $0 < M_{ij} < 1$ and $\vartheta(M) \geq \sigma$, for some constant $\sigma > 0$. Let $R \in \mathbb{R}^{n \times n}$ be a perturbation matrix, such that its entries $|R_{ij}| < \epsilon/2$, $\forall (i, j)$, and $0 < \epsilon < \sigma^2/3n^3$. Let $h_{\min}$ be a smallest magnitude of the altitude of the simplex corresponding to the matrix $M + R$. One has that*

$$h_{\min} \geq \sqrt{\sigma^2 - 6n^3\epsilon}. \tag{34}$$

**Proof of Lemma 18.** Denote $\Delta$ as the simplex corresponding to $M = [x^1, ..., x^n]$, $\Delta_i$ as the facet opposite the vertex $x^i$, and $h_i(\Delta)$ is the height of simplex w.r.t. the vertex $x^i$. Denote $\text{Vol}_k(C)$ as the $k$-dimensional content of $C \subset \mathbf{E}^{n-1}$, where $\dim(C) = k$. Using simple calculus, one has that

$$h_i(\Delta) = \frac{1}{n-1} \frac{\text{Vol}_{n-1}(\Delta)}{\text{Vol}_{n-2}(\Delta_i)}, \tag{35}$$

We also denote $\hat{\Delta}$ as the perturbed simplex corresponding to $M + R$ and $\hat{\Delta}_i$ as the facet opposite the to the perturbation of $x^i$.

we bound the height $h_i(\hat{\Delta})$ for all $i \in [n-1]$. For the height $h_n(\hat{\Delta})$, one can apply similar reasoning. Let define the coordinate matrix and the pertubation matrix w.r.t $x^n$ as follows

$$V := \text{coM}(M, n), \quad U := \text{coM}(R, n); \tag{36}$$

We have that $|U_{ij}| < \epsilon$, $\forall i, j$. By the definition of width $\vartheta(M)$, we have that

$$\sigma_{n-1}(V) \geq \vartheta(M) \geq \sigma. \tag{37}$$

Let define the Gram matrix and perturbed Gram matrix as follows

$$G := V^\top V$$
$$\hat{G} := (V + U)^\top (V + U). \tag{38}$$

One has that,

$$\hat{G} - G = V^\top U + U^\top V + U^\top U := \overline{U}.$$

One has that $\overline{U}_{ij} \leq \bar{\epsilon} := 3n\epsilon$, as $|V_{ij}| < 1$ and $|U_{ij}| < \epsilon < 1$. We also has that $\|\overline{U}\|_2 \leq \|\overline{U}\|_F \leq n\bar{\epsilon}$.

**First step.** we bound the quantity $\frac{|\det(G+\overline{U}) - \det(G)|}{|\det(G)|}$. By [12]'s Corollary 2.14, one has that

$$\frac{|\det(G + \overline{U}) - \det(G)|}{|\det(G)|} \leq \left(1 + \frac{\|\overline{U}\|_2}{\sigma_{n-1}(G)}\right)^{n-1} - 1 \leq \left(1 + \frac{n\bar{\epsilon}}{\sigma^2}\right)^n - 1. \tag{39}$$

As $(1 + z)^n \leq \frac{1}{1-nz}$ when $z \in (0, \frac{1}{n})$ and $n > 0$. One has that

$$\frac{|\det(G + \overline{U}) - \det(G)|}{|\det(G)|} \leq \frac{n^2\bar{\epsilon}}{\sigma^2 - n^2\bar{\epsilon}}, \tag{40}$$

when $\bar{\epsilon} \leq \frac{\sigma^2}{n^2}$, or $\epsilon \leq \frac{\sigma^2}{3n^3}$. Let us define $k := \frac{\sigma^2 - n^2\bar{\epsilon}}{n^2\bar{\epsilon}}$, one has that

$$\frac{|\det(G + \overline{U}) - \det(G)|}{|\det(G)|} \leq \frac{1}{k}. \tag{41}$$

It means that

$$\det(G + \overline{U}) \geq \left(1 - \frac{1}{k}\right) \det(G) \tag{42}$$

**Second step.** we bound the change in content of the $i^{\text{th}}$ facets of the simplex, for $i \in [n-1]$. Consider the facet that is opposite to the vertex $x^i$, and denote $V(i)$, $U_i$ as the sub-matrices of $V$, $U$ by removing $i^{\text{th}}$ column. Denote the Gram matrix

$$G(i) := V(i)^\top V(i)$$
$$\hat{G}(i) := (V(i) + U(i))^\top (V(i) + U(i)) \tag{43}$$

Note that one can obtain $G(i)$, $\hat{G}(i)$ and by removing $i^{\text{th}}$ row and column of $G(i)$, $\hat{G}(i)$ respectively. Denote $\overline{U}(i) :=$

$\hat{G}(i) - G(i)$, we has that all entries of $\overline{U}(i)$ smaller than $\bar{\epsilon}$.

Moreover, by Singular Value Interlacing Theorem, one has that

$$\sigma_1(G) \geq \sigma_1(G(i)) \geq \sigma_2(G) \geq \sigma_2(G(i)) \geq \cdots \geq \sigma_{n-2}(G(i)) \geq \sigma_{n-2}(G(i)) \geq \sigma_{n-1}(G). \tag{44}$$

Similarly, one has that

$$\frac{|\det(G(i) + \overline{U}(i)) - \det(G(i))|}{|\det(G(i))|} \leq \left(1 + \frac{\|\overline{U}(i)\|_2}{\sigma_{n-2}(G(i))}\right)^{n-2} - 1 \leq \left(1 + \frac{n\bar{\epsilon}}{\sigma^2}\right)^n - 1 \leq \frac{1}{k}. \tag{45}$$

It means that

$$\det(G(i) + \overline{U}(i)) \leq \left(1 + \frac{1}{k}\right)\det(G(i)). \tag{46}$$

**Third step.** We bound the height $h_i$ corresponding to the vertices $x^i$ in this step. For $i \in [n-1]$ one has that

$$\text{Vol}_d(\hat{\Delta}) = \frac{1}{(n-1)!}\sqrt{\det(G + \overline{U})}.$$
$$\text{Vol}_{d-1}(\hat{\Delta}_i) = \frac{1}{(n-2)!}\sqrt{\det(G(i) + \overline{U}(i))}. \tag{47}$$

Furthermore, by the Eigenvalue Interlacing Theorem, we have $\det(G)/\det(G_i) \geq \sigma_{n-1}(G) \geq \sigma^2$. Putting things together, one has that

$$h_i(\hat{\Delta}) = \frac{1}{n-1}\frac{\text{Vol}_{n-1}(\hat{\Delta})}{\text{Vol}_{n-2}(\hat{\Delta}_i)} = \sqrt{\frac{\det(G + \overline{U})}{\det(G(i) + \overline{U}(i))}} \geq \sqrt{\frac{k-1}{k+1}\frac{\det(G)}{\det(G(i))}} \geq \sigma\sqrt{\frac{\sigma^2 - 6n^3\epsilon}{\sigma^2}} \tag{48}$$

We note that, the above holds true for $i \in [n-1]$.

**Fourth Step.** Now, we bound the height corresponding to the vertex $x^n$. We can define the coordination matrix and pertubation matrix w.r.t $x^1$ as follows.

$$V' = \text{coM}(M, 1), \quad U' = \text{coM}(R, 1). \tag{49}$$

Note that, by the definition of the width, we have that

$$\sigma_{n-1}(V') \geq \vartheta(M) \geq \sigma; \tag{50}$$

and also, $|U'_{ij}| \leq \epsilon$. Similarly, applying Steps 1-3, we also have that

$$h_n(\hat{\Delta}) \geq \sqrt{\sigma^2 - 6n^3\epsilon}$$

Therefore, $h_i(\hat{\Delta}) \geq \sqrt{\sigma^2 - 6n^3\epsilon}$ holds true for all $i \in [n]$. We conclude that

$$h_{\min} \geq \sqrt{\sigma^2 - 6n^3\epsilon}, \tag{51}$$

whenever, $\epsilon \leq \frac{\sigma^2}{3n^3}$. $\qquad\square$

Lemma 18 guarantees that if the width of the ground truth simplex is relatively large compared to the diameter of the confidence set, then the heights of the estimated simplex are also large. We now provide an example of a collection of permutation orders corresponding to a set of vertices as follows. Let $s_i := (i, i+1)$ denote the *adjacent transposition* between $i$ and $i+1$.

**Proposition 19.** *Fix any $\omega \in \mathfrak{S}_n$, consider the collection of permutation $\mathcal{P} = \{\omega, \omega s_1, \ldots, \omega s_{n-1}\}$ and matrix $M = [\phi^{\omega'}]_{\omega' \in \mathcal{P}}$. The width of the simplex that corresponds to $M$, is upper bounded as $\vartheta(M) \geq 0.5\varsigma n^{-3/2}$.*

The vertex set in Proposition 19 comprises one vertex and its $(n-1)$ adjacent vertices. Combining Lemma 18, Proposition 19 with the stopping condition provided by Theorem 9, we now can guarantee the sample complexity of our algorithm as Theorem 10.

**Proof of Theorem 10.** Let choose the collection $\mathcal{P}$ as Proposition 19. Note that $|\mathcal{P}| = n$. Denote $\epsilon_0 := 2\max_{p\in[n]}\text{diam}(\mathcal{C}_p)$. For any $\omega_p \in \mathcal{P}$, define the event

$$\mathcal{E} = \{\phi^{\omega_p} \in \mathcal{C}_p, \forall p \in [n]\}.$$

By the construction of the confidence set, we guarantee that $\mathcal{E}$ happen with probability at least $1 - n^2\delta$.

Consider $p \in [n]$, for any $q \in [n] \setminus p$, let $x^q$ be the projection of $\phi^{\omega_q}$ onto $H_p$, and $x^p := \arg\min_{p\in\mathcal{C}_p} \mathcal{D}(x, H_p)$. We have that

$$\mathcal{D}(x^k, \phi^{\omega_k}) \leq \epsilon_0, \ \forall k \in [n].$$

We need to bound $\mathcal{D}(x^p, H_p)$ by bounding the minimum height of simplex $\text{Conv}\left(\{x^p\}_{p\in[n]}\right)$, which is a pertubation of $\text{Conv}\left(\{\phi^{\omega_p}\}_{p\in[n]}\right)$.

Define matrix $M = [\phi^{\omega_p}]_{p\in[n]}$, and $\hat{M} = [x^p]_{p\in[n]}$. Let $R := M - \hat{M}$ be the perturbation matrix, one has that $R_{ij} \leq \epsilon_0, \ \forall(i,j)$. By Lemma 18, we have that

$$\mathcal{D}(x^p, H_p) \geq \sqrt{\sigma^2 - 12n^3\epsilon_0} \tag{52}$$

Therefore, for $\mathcal{D}(x^p, H_p) \geq n\epsilon_0$ holds , it is sufficient to provide the condition for $\sigma$ such that

$$\sqrt{\sigma^2 - 12n^3\epsilon_0} \geq n\epsilon_0. \tag{53}$$

Assuming that $\epsilon_0 < 1$, for the condition of Lemma 18 and the above inequality to hold, it is sufficient to choose

$$\epsilon_0 = \frac{\sigma^2}{13n^3}.$$

Now, we calculate the upper bound for sample needed. At epoch $K$, we have that

$$\epsilon_0 = 2\text{diam}(\mathcal{C}_p) \geq 4\sqrt{\frac{2n\log(\delta^{-1})}{K}}$$
$$\sigma = \frac{n\varsigma}{c_W}. \tag{54}$$

Then we have $K = O\left(\frac{n^{13}\log(n\delta^{-1}\varsigma^{-1})}{\varsigma^4}\right)$. As each phase, there are at most $n^2$ queries, then the total number of sample needed is

$$T = O\left(\frac{n^{15}\log(n\delta^{-1}\varsigma^{-1})}{\varsigma^4}\right) \tag{55}$$

for the algorithm to return a common point, with probability of at least $1 - \delta$. □

While the choice of vertices in Proposition 19 achieves polynomial sample complexity, the width of the simplex decreases with dimension growth, hindering its sub-optimality. An alternative choice of vertices is those corresponding to cyclic permutation, denoted as $\mathfrak{C}_n \subset \mathfrak{S}_n$, which have a larger width in large subsets of strictly convex games (as observed in simulations) but can be difficult to verify in the worst case. We refer readers to Appendix D.3 for the detail simulation and discussion on the choice of set of $n$ vertices. Based on this observation, we achieve the sample complexity which better dependence on $n$ as follows.

**Theorem 20.** *Suppose Assumption 3 holds. Let $\mathcal{P} = \mathfrak{S}_n$ the collection of cyclic permutations, and denote the coordinate matrix of the corresponding vertices as $W$. Assume that the width of the simplex $\vartheta(W) \geq \frac{n\varsigma}{c_W}$ for some $c_W > 0$. Then, for*

*any $\delta \in [0,1]$, if number of samples is*

$$T = O\left(\frac{n^5 c_W^4 \log(n c_W \delta^{-1} \varsigma^{-1})}{\varsigma^4}\right), \tag{56}$$

*the* `Common-Points-Picking` *algorithm returns a point in E-Core with probability at least* $1 - \delta$.

*Proof of Theorem 20.* The proof is identical to that of Theorem 10, with the width of the simplex bounded by $\vartheta(W) \geq \frac{n\varsigma}{c_W}$. $\qquad\square$

## D. Convex Games

### D.1. E-Core of convex games and Generalised Permutahedra

Formulating the coordinates of the vertices of the core can be achieved using the connection between the core of a convex game and the generalised permutahedron. There is an equivalence between generalised permutahedra and polymatroids; it was also shown in [20] that the core of each convex game is a generalised permutahedron.

For any $\omega \in \mathfrak{S}_n$, let $\mathbf{I}^\omega = (\omega(1), ..., \omega(n))$. The $n$-permutahedron is defined as $\mathrm{Conv}\left(\{\mathbf{I}^\omega \mid \omega \in \mathfrak{S}_n\}\right)$. A generalised permutahedron can be defined as a deformation of the permutahedron, that is, a polytope obtained by moving the vertices of the usual permutohedron so that the directions of all edges are preserved [17]. Formally, the edge of the core corresponding to adjacent vertices $\phi^\omega$, $\phi^{\omega s_i}$ can be written as

$$\phi^\omega - \phi^{\omega s_i} = k_{\omega,i}(e_{\omega(i)} - e_{\omega(i+1)}), \tag{57}$$

Where, $k_{\omega,i} \geq 0$, and $e_1, \ldots e_n$ are the coordinate vectors in $\mathbb{R}^n$. If the game is $\varsigma$-strictly convex, $k_{\omega,i} > \varsigma$.

### D.2. Proof of Proposition 19

We utilise the formulation of edges of the generalized permutahedron as described in Subsection D.1 to calculate the matrix of coordinates for the vertices of E-Core. Based on the matrix of coordinates, we now state the proof of Proposition 19.

*Proof of Proposition 19.* As the set of vertices is $\phi^\omega$ and its $n-1$ neighbors, there are only two cases to consider. First, we need to consider the matrix created by using $\phi_\omega$ as the reference, that is $\mathrm{coM}(M,1)$. As the neighbors have the same roles, bounding the width of the matrices using any neighbor as a reference point can be done identically. Therefore, we will prove the theorem for $\mathrm{coM}(M,2)$, and the proof for $\mathrm{coM}(M,i)$, $i \neq 1$ can be done in the same manner. Let us denote

$$V = \mathrm{coM}(M,1) = \begin{bmatrix} c_1 & 0 & 0 & \cdots & 0 & 0 \\ -c_1 & c_2 & 0 & \cdots & 0 & 0 \\ 0 & -c_2 & c_3 & \cdots & 0 & 0 \\ \vdots & \vdots & \vdots & \ddots & \vdots & \vdots \\ \vdots & \vdots & \vdots & \ddots & \vdots & \vdots \\ 0 & 0 & 0 & \cdots & -c_{n-2} & c_{n-1} \\ 0 & 0 & 0 & \cdots & 0 & -c_{n-1} \end{bmatrix} \in \mathbb{R}^{n \times (n-1)}, \tag{58}$$

$$U = \mathrm{coM}(M,2) = \begin{bmatrix} -c_1 & -c_1 & -c_1 & -c_1 & \cdots & -c_1 & -c_1 \\ c_1 & c_1+c_2 & c_1 & c_1 & \cdots & c_1 & c_1 \\ 0 & -c_2 & c_3 & 0 & \cdots & 0 & 0 \\ 0 & 0 & -c_3 & c_4 & \cdots & 0 & 0 \\ \vdots & \vdots & \vdots & \vdots & \ddots & \vdots & \vdots \\ \vdots & \vdots & \vdots & \vdots & \ddots & \vdots & \vdots \\ 0 & 0 & 0 & 0 & \cdots & -c_{n-2} & c_{n-1} \\ 0 & 0 & 0 & 0 & \cdots & 0 & -c_{n-1} \end{bmatrix} \in \mathbb{R}^{n \times (n-1)}, \tag{59}$$

in which each $c_i > \varsigma$.

We will exploit the following norm inequality in the proof. For any $A_1, \ldots, A_n \in \mathbb{R}$, we use the following inequality (norm 2 vs. norm 1 of vectors)

$$\sum_{i=1}^{n} A_i^2 \geq \frac{(\sum_{i=1}^{n} A_i)^2}{n} \tag{60}$$

**Consider V.** Consider a unit vector $x = (x_1, ..., x_{n-1})$. We have

$$V x = \begin{bmatrix} c_1 x_1 \\ -c_1 x_1 + c_2 x_2 \\ -c_2 x_2 + c_3 x_3 \\ \cdots \\ -c_{n-2} x_{n-2} + c_{n-1} x_{n-1} \\ -c_{n-1} x_{n-1} \end{bmatrix} \tag{61}$$

Applying the Ineq. (60) for $A_1 = c_1 x_1$, $A_2 = -c_1 x_1 + c_2 x_2$, $A_{n-1} = -c_{n-2} x_{n-2} + c_{n-1} x_{n-1}$, $A_n = -c_{n-1} x_{n-1}$ gives

$$\|V x\|^2 \geq \frac{c_1^2 x_1^2}{n} \geq \frac{\varsigma^2 x_1^2}{n};$$

$$\|V x\|^2 \geq c_1^2 x_1^2 + (-c_1 x_1 + c_2 x_2)^2 \geq \frac{c_2^2 x_2^2}{n} \geq \frac{\varsigma^2 x_2^2}{n}; \tag{62}$$

$$\cdots$$

$$\|V x\|^2 \geq \frac{\varsigma^2 x_{n-1}^2}{n}.$$

Therefore,

$$n \|V x\|^2 \geq \frac{\varsigma^2 (x_1^2 + \cdots + x_{n-1}^2)}{n} = \frac{\varsigma^2}{n} \tag{63}$$

Therefore $\|V x\| \geq \varsigma/n$, hence $\sigma_{n-1}(V) \geq \varsigma/n$.

**Consider U.** Similarly, consider a unit vector $x = (x_1, ..., x_{n-1})$. We have

$$U x = \begin{bmatrix} -c_1(x_1 + x_2 + ... + x_{n-1}) \\ c_1(x_1 + x_2 + ... + x_{n-1}) + c_2 x_2 \\ -c_2 x_2 + c_3 x_3 \\ -c_3 x_3 + c_4 x_4 \\ \cdots \\ -c_{n-2} x_{n-2} + c_{n-1} x_{n-1} \\ -c_{n-1} x_{n-1} \end{bmatrix} \tag{64}$$

Applying the Ineq. (60) for $A_1 = c_1(x_1 + x_2 + ... + x_{n-1})$, $A_2 = c_1(x_1 + x_2 + ... + x_{n-1}) + c_2 x_2$, $A_3 = -c_2 x_2 + c_3 x_3$, $A_4 = -c_3 x_3 + c_4 x_4$, $\ldots$, $A_{n-1} = -c_{n-2} x_{n-2} + c_{n-1} x_{n-1}$, $A_n = -c_{n-1} x_{n-1}$ gives

Note that

$$\|U x\|^2 \geq \frac{\varsigma^2 (x_1 + x_2 + ... + x_{n-1})^2}{n};$$

$$\|U x\|^2 \geq c_1^2 (x_1 + x_2 + ... + x_{n-1})^2 + (c_1(x_1 + x_2 + ... + x_{n-1}) + c_2 x_2)^2 \geq \frac{c_2^2 x_2^2}{n} \geq \frac{\varsigma^2 x_2^2}{n};$$

$$\|U x\|^2 \geq c_1^2 (x_1 + x_2 + ... + x_{n-1})^2 + (c_1(x_1 + x_2 + ... + x_{n-1}) + c_2 x_2)^2 + (-c_2 x_2 + c_3 x_3)^2 \geq \frac{\varsigma^2 x_3^2}{n}; \tag{65}$$

$$\cdots$$

$$\|U x\|^2 \geq \frac{\varsigma^2 x_{n-1}^2}{n}$$

Therefore, we also have

$$n\|Ux\|^2 \geq \frac{\varsigma^2((x_1 + x_2 + ... + x_{n-1})^2 + x_2^2 + ... + x_{n-1}^2)}{n} \geq \frac{\varsigma^2 x_1^2}{n^2} \tag{66}$$

From that, we have that

$$2n\|Ux\|^2 \geq \varsigma^2 \frac{x_1^2}{n^2} + \frac{x_2^2}{n} + \cdots + \frac{x_{n-1}^2}{n} \geq \frac{x_1^2 + ... + x_{n-1}^2}{n^2} = \frac{\varsigma^2}{n^2}, \text{ as } \|x\| = 1 \tag{67}$$

That is, $\|Ux\| \geq \frac{\varsigma^2}{\sqrt{2n^3}}$. Therefore, $\sigma_{n-1}(U) \geq \frac{\varsigma^2}{\sqrt{2n^3}}$.

Therefore, we have that $\vartheta(M) > \frac{\varsigma^2}{\sqrt{2n^3}}$. □

### D.3. Alternative choice of $n$ vertices of E-Core

In this subsection, we provide an alternative choice of vertices rather than that in Proposition 19. Recall that, with the choice of vertices in Proposition 19, the lower bound for the width of the simplex diminishes when the dimension increases. This leads to a large dependence of the sample complexity on $n$. To mitigate this, we investigate other choices of $n$ vertices. To see this, we first recall the equivalence between E-Core and generalized permutahedra as explained in Subsection D.1.

However, even in the case of a simple permutahedron, if the set of vertices is not carefully chosen, the width of their convex can be proportionally small w.r.t. $n$, as demonstrated in the next proposition. In particular, the same choice of vertices as in 19 results in the simplex with diminishing width as follows.

**Proposition 21.** *Consider a permutahedron, fix $\omega \in \mathfrak{S}_n$, consider the matrix $W = [\phi^\omega, \mathbf{I}^{\omega s_1}, \mathbf{I}^{\omega s_2}, \ldots, \mathbf{I}^{\omega s_{n-1}}]$. The width of the simplex that corresponds to $M$, is upper bounded as follows:*

$$\vartheta(M) \leq \frac{3}{n}. \tag{68}$$

*Proof.* The coordinate matrix w.r.t. $\phi^\omega$, that is, $\mathrm{coM}(M, 1)$ can be written as follows.

$$V = \begin{bmatrix} 1 & 0 & 0 & \cdots & 0 & 0 \\ -1 & 1 & 0 & \cdots & 0 & 0 \\ 0 & -1 & 1 & \cdots & 0 & 0 \\ \vdots & \vdots & \vdots & \ddots & \vdots & \vdots \\ \vdots & \vdots & \vdots & \ddots & \vdots & \vdots \\ 0 & 0 & 0 & \cdots & -1 & 1 \\ 0 & 0 & 0 & \cdots & 0 & -1 \end{bmatrix} \in \mathbb{R}^{n \times (n-1)} \tag{69}$$

Therefore, the Gram matrix is

$$G := V^\top V = \begin{bmatrix} 2 & -1 & 0 & 0 & 0 & \cdots & 0 & 0 & 0 \\ -1 & 2 & -1 & 0 & 0 & \cdots & 0 & 0 & 0 \\ 0 & -1 & 2 & -1 & 0 & \cdots & 0 & 0 & 0 \\ \vdots & \vdots & \vdots & \vdots & \vdots & \ddots & \vdots & \vdots & \vdots \\ \vdots & \vdots & \vdots & \vdots & \vdots & \ddots & \vdots & \vdots & \vdots \\ \vdots & \vdots & \vdots & \vdots & \vdots & \ddots & \vdots & \vdots & \vdots \\ \vdots & \vdots & \vdots & \vdots & \vdots & \ddots & \vdots & \vdots & \vdots \\ 0 & 0 & 0 & \cdots & 0 & 0 & -1 & 2 & -1 \\ 0 & 0 & 0 & \cdots & 0 & 0 & 0 & -1 & 2 \end{bmatrix} \in \mathbb{R}^{(n-1) \times (n-1)}. \tag{70}$$

Note that $G$ is a tridiagonal matrix and also Toeplitz matrix, therefore, its minimum eigenvalues has closed form as follows

$$\lambda_{n-1}(G) = 2 + 2\cos\left(\frac{(n-1)\pi}{n}\right) = 2\sin^2\left(\frac{\pi}{2n}\right) \le \frac{5}{n^2}; \tag{71}$$

as $\left|\sin\left(\frac{\pi}{2n}\right)\right| \le \frac{\pi}{2n}$. Therefore, $\vartheta(M) \le \sigma_{n-1}(V) = \sqrt{\lambda_{n-1}(G)} \le \frac{3}{n}$. $\qquad\square$

Proposition 21 highlights the challenge of selecting a set of vertices such that the width does not contract with the increasing dimension, even in the case of a simple permutahedron. Denote $\mathfrak{C}_n \subset \mathfrak{S}_n$ as the group of cyclic permutations of length $n$. One potential candidate for such a set of vertices is the collection corresponding to cyclic permutations $\mathfrak{C}_n$, as described in the next proposition.

**Proposition 22.** *Consider the matrix $\overline{W} = [\mathbf{I}^\omega]_{\omega\in\mathfrak{C}_n}$. We have that*

$$\vartheta(\overline{W}) \ge \frac{n}{2}. \tag{72}$$

*Proof.* The form of matrix $\overline{W}$ is as follows

$$\overline{W} = \begin{bmatrix} 1 & n & n-1 & \dots & 2 \\ 2 & 1 & n & \dots & 3 \\ 3 & 2 & 1 & \dots & 4 \\ \vdots & \vdots & \vdots & \vdots & \vdots \\ n-1 & n-2 & n-3 & \dots & n-1 \\ n & n-1 & n-2 & \dots & 1 \end{bmatrix}. \tag{73}$$

The coordinate matrix w.r.t. the first column is as follows

$$V = \mathrm{coM}(\overline{W}, 1) = \begin{bmatrix} n-1 & n-2 & \dots & 1 \\ -1 & n-2 & \dots & 1 \\ -1 & -2 & \dots & 1 \\ \vdots & \vdots & \ddots & \vdots \\ -1 & -2 & \dots & 1 \\ -1 & -2 & \dots & -(n-1) \end{bmatrix}. \tag{74}$$

Let $u \in \mathbb{R}^{n-1}$ be any unit vector, and let $z = Vu \in \mathbb{R}^n$. We have that

$$z_i - z_{i+1} = nu_i. \tag{75}$$

Let us consider

$$\begin{aligned} 4\|z\|^2 &= 4z_1^2 + 4z_2^2 + \dots + 4z_n^2 \\ &= 2z_1^2 + [(z_1 + z_2)^2 + (z_1 - z_2)^2] + [(z_2 + z_3)^2 + (z_2 - z_3)^2] \\ &\quad + \dots + [(z_{n-1} + z_n)^2 + (z_{n-1} - z_n)^2] + 2z_n^2 \\ &\ge (z_1 - z_2)^2 + (z_2 - z_3)^2 + \dots + (z_{n-1} - z_n)^2 \\ &= n^2(u_1^2 + u_2^2 + \dots + u_{n-1}^2) = n^2. \end{aligned} \tag{76}$$

Therefore, we have that

$$\sigma_{n-1}(V) = \min_{u:\|u\|=1} \sqrt{\frac{\|Vu\|^2}{\|u\|^2}} \ge \frac{n}{2}. \tag{77}$$

It is straightforward that if one takes any column of $\overline{W}$ as a reference column, the resulting coordinate matrices have identical singular values. In particular, for any $i, j \in [n]$

$$\mathrm{coM}(\overline{W}, i) = P \cdot \mathrm{coM}(\overline{W}, j),$$

where $P$ is a permutation matrix, thus, their singular values are identical. Therefore, we have that

$$\vartheta(\overline{W}) \geq \frac{n}{2}.$$

$\square$

As a result, the set of vertices corresponding to cyclic permutations is a sensible choice. In case of a generalised permutahedron, let us define

$$W := [\phi^\omega]_{\omega \in \mathfrak{C}_n}. \tag{78}$$

As generalised permutahedra are deformations of the permutahedron, we expect that $\vartheta(W)$ is reasonably large for a broad class of strictly convex games. In particular, we consider the class of strictly convex games in which the width $\vartheta(W)$ is lower bounded, as in the following assumption:

**Assumption 23.** *The width of the simplex that corresponds to $W$ in* (78) *is bounded as follows*

$$\vartheta(W) \geq \frac{n\varsigma}{c_W}, \tag{79}$$

*for some constant $c_W > 0$.*

These parameters will eventually play a crucial role in determining the number of samples required using this choice of $n$ permutation orders. Although proving an exact upper bound for $c_W$ in all strictly convex games is challenging, we conjecture that $c_W$ is relatively small in a large subset of the games.

To investigate Assumption 23, we conducted a simulation to compute the constant $c_W$ of the minimum singular value $\sigma_{n-1}(M)$. For each case where $n$ takes values of $(10, \ 50, \ 100, \ 150, \ 200, \ 300, \ 500, \ 1000)$, the simulation consisted of 20000 game trials with $\varsigma = 0.1/n$. As depicted in Figure 1, the values of $c_W$ tend to be relatively small and highly concentrated within the interval $(0, 30)$. This observation suggests that for most cases of strictly convex games, $c_W$ remains reasonably small. Consequently, our algorithm exhibits relatively low sample complexity.

For each case where $n$ takes values of $(10, \ 50, \ 100, \ 150, \ 200, \ 300, \ 500, \ 1000)$, the simulation consisted of 20000 game trials with $\varsigma = 0.1/n$. As depicted in Figure 1, the values of $c_W$ tend to be relatively small and highly concentrated within the interval $(0, 30)$. This observation suggests that for most cases of strictly convex games, $c_W$ remains reasonably small. The results indicate that $c_W$ tends to be relatively small with high probability, and does not depend on the value of $n$.

# E. Further Discussions

## E.1. Comparison with Pantazis *et al.* [16]

While the algorithm in [16] is proposed for general cooperative games and conceptually applicable to the class of strictly convex games, we argue that their algorithm is not statistically and computationally efficient when applied to strictly convex games, due to the absence of a specific mechanism to leverage the supermodular structure of the expected reward function. In particular, firstly, we argue that without any modification and with bandit feedback, their algorithm would require a minimum of $\Omega(2^n)$ samples. Secondly, although we believe the framework of [16] could be conceptually applied to strict convex games, significant non-trivial modifications may be necessary to leverage the supermodular structure of the mean reward function.

**Appplying [16] to strictly convex games without any modifications.** We first briefly outline their algorithmic framework. In this paper, the authors assume that each coalition $S \subset N$ has access to a number of samples, denoted as $t_S > 1$. For each coalition $S$, the empirical mean is denoted as $\overline{\mu}_{t_S}(S)$, and a confidence set for the given mean reward is constructed, denoted as,

$$\mathcal{C}(\mu(S)) = \left\{ \hat{\mu}(S) \in [0, \ 1] \mid |\hat{\mu}(S) - \overline{\mu}_{t_S}(S)| \leq \varepsilon_{t_S} \right\}, \text{ for some } \varepsilon_{t_S} > 0 \,.$$

We note that while the algorithm in [16] constructs the confidence set using Wasserstein distance, in the case of distributions with bounded support, we can simplify it by using the mean reward difference. After constructing the confidence set for the

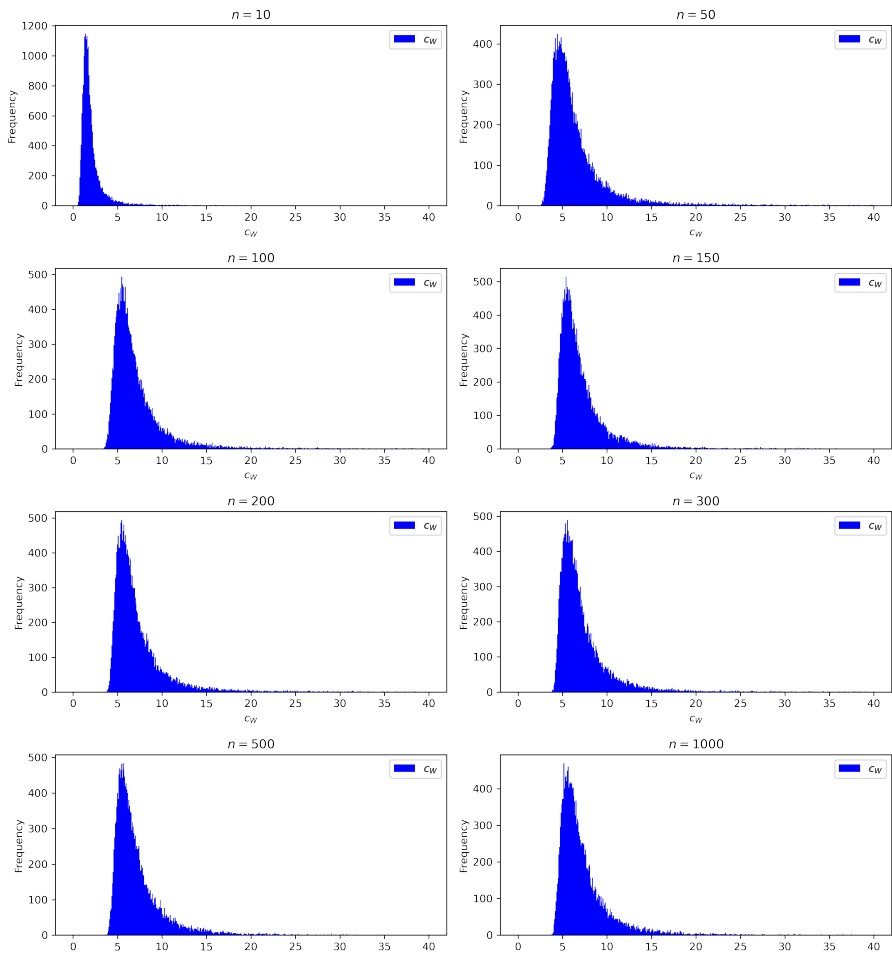

*Figure 1.* $c_W$ with $n \in \{10, \ 50, \ 100, \ 150, \ 200, \ 300, \ 500, \ 1000\}$, $\varsigma = \frac{0.1}{n}$, and 20000 trials

mean reward of each coalition, the algorithm solves the following robust optimization problem:

$$\min_{x \in \mathbb{R}^n} \|x\|_2^2$$

$$\text{s.t. } x(N) = \mu(N)$$

$$x(S) \geq \sup(\mathcal{C}(\mu(S))), \quad \forall S \subset N.$$

That is, finding the stable allocation for the worst-case scenario within the confidence sets. It is clear that when directly applying this framework to the bandit setting, each coalition must be queried at least once, that is $t_S > 1$. This inevitably leads to a complexity of $\Omega(2^n)$ samples, regardless of the sampling scheme one employs. In term of computation, with $2^n - 2$ confidence sets for all coalitions $S \subset N$, tabular representation of the confidence set incurs extreme computational cost.

**Significant modifications required for [16].**    As described above, the algorithm in [16] suffers from $2^n$ sample complexity, and the main reason is because it requires constructing confidence sets for the mean reward for all coalitions $S \subset N$. As such, if we want to apply their algorithm efficiently to the bandit setting, we need to address this limitation.

To do so, one may need to develop an approach to design a confidence set for a specific class of strictly convex games. For instance, we can consider the following approach: Given historical data, instead of writing a confidence set for each

individual coalition, let us define a confidence set for the mean reward function as follows:

$$\mathcal{C}(\mu) = \left\{ \hat{\mu} : 2^N \to [0,\ 1] \mid \hat{\mu} \in [\mathcal{C}(\mu(S))]_{S \subset N},\ \hat{\mu} \text{ is strictly supermodular} \right\}; \tag{80}$$

where the confidence set $\mathcal{C}(\mu(S))$ could potentially be $[0, 1]$ for some coalition $S$, as there is no data available for these coalitions. Let $\mathrm{Core}(\hat{\mu})$ be the core with respect to the reward function $\hat{\mu}$. We propose a generalization of the framework from the robust optimization problem to adapt to the structure of the game as follows.

$$\min_{x \in \mathbb{R}^n} \|x\|_2^2$$
$$\text{s.t. } x(N) = \mu(N) \tag{81}$$
$$x \in \bigcap_{\hat{\mu} \in \mathcal{C}(\mu)} \mathrm{Core}(\hat{\mu}).$$

That is, we find a stable allocation $x$ for every possible supermodular function within the confidence set of the reward function.

However, implementing and analyzing this approach may pose significant challenges. The first challenge lies in constructing a tight confidence set $[\mathcal{C}(\mu(S))]_{S \subset N}$ such that all functions within this collection are strictly supermodular. We are not aware of a method to explicitly construct $[\mathcal{C}(\mu(S))]_{S \subset N}$ containing only strictly supermodular functions, and we believe this set could potentially be very complicated. To illustrate, consider the scenario where we have samples from two coalitions, $\{1\}$ and $\{1, 2\}$, with the following empirical means:

$$\overline{\mu}(\{1\}) = 0.11; \quad \overline{\mu}(\{1, 2\}) = 0.1$$

This situation might occurs when the number of samples is insufficient. In such cases, regardless of the value chosen for the remaining coalition rewards in the function $\overline{\mu}(S)$, $\overline{\mu}(S)$ is not supermodular (as $\{1\} \subset \{1, 2\}$, yet $\overline{\mu}(1) > \overline{\mu}(1, 2)$). Consequently, either the confidence set $\mathcal{C}(\mu(1))$ or $\mathcal{C}(\mu(1, 2))$ does not contain the empirical mean reward, indicating the highly complicated shape of the confidence set.

The second challenge is that while computing a stable allocation for a given supermodular reward function $\hat{\mu}$ is a straightforward task, computing a stable allocation for all supermodular reward functions in the confidence set $\mathcal{C}(\mu)$ in a computationally efficient way is an open problem, to the best of our knowledge.

The discussion above also highlights the key difference between our work and that of [16]: Instead of explicitly constructing the confidence set of the expected mean reward function to integrate the supermodular structure for computing a stable allocation, which might be a sophisticated task, we directly exploit the geometry of the core of strictly convex games. Specifically, in strictly convex games, each vertex of the core corresponds to a marginal vector with respect to some permutation orders. Given that one can construct the confidence set of marginal vectors easily, our method is conceptually and computationally simpler. However, we believe that adopting the more general framework of robust optimization as presented in [16] is a very interesting, but non-trivial, direction, and we leave it for future work.

