# OpenReview forum: "Learning Stable Allocations of Strictly Convex Stochastic Cooperative Games"
_ICML.cc/2024/Workshop/Agentic_Markets — Agentic Markets @ ICML'24 Poster_

### Official Review · Reviewer_fAz7 · 2024-06-12
**Common-Points-Picking Algorithm for Learning the Expected Core in Stochastic Cooperative Games**

**Rating:** 7
**Confidence:** 1

**Review:**

This paper introduces a new algorithm, Common-Points-Picking, to learning the expected core in strictly convex stochastic cooperative games. This algorithm helps to learn the expected core using a polynomial number of samples, even when the exact reward distribution is unknown and only partial information is available.

**Pros:**
- Strong theoretical foundation and robust mathematical analysis.
- A nice comparison with a similar work where authors highlight their algorithm’s simplicity, both conceptually and computationally, compared to the other work.
- The abstract effectively summarizes the problem, proposed solution, and contributions, clearly communicating the paper's purpose.
- The paper fits well within the scope of ICML and the workshop. It addresses central topics in cooperative AI systems, including reward allocation and stability.
- The comparison with Pantazis et al. [1] in Appendix E provides room for discussion and interactive engagement during the workshop.

**Cons:**
- More intuitive illustrations and practical examples could enhance understanding of the theoretical concepts and the effectiveness of the algorithms.

[1] G. Pantazis, B. Franci, S. Grammatico, and K. Margel- los. Distributionally robust stability of payoff alloca- tions in stochastic coalitional games. arXiv preprint arXiv:2304.01786, 2023.

---

### Official Review · Reviewer_N81J · 2024-06-14
**Finds the expected code without requiring knowledge of the reward distribution in cooperative games**

**Rating:** 7
**Confidence:** 3

**Review:**

## Summary
Deals with the problem of learning stable reward allocations in cooperative games where the underlying reward distribution is stochastic/unknown. Cooperative games aim to find stable reward allocations such that none of the agents break away from coalitions. Finding this ‘core’ has normally required knowledge of the reward distribution. This paper presents the first step of finding the expected core without perfect knowledge of the reward distribution by sampling from the environment to obtain rewards (a polynomial number of reward samples are required).

## Strengths
- Common Points Picking Algorithm: Learns the expected core using a polynomial number of reward samples. Querying vertex estimates and using their geometric arrangement to estimate the vertices of the expected core simplex is novel.
- Works in stochastic cooperative games/games where the reward distribution is unknown.
- Extensions to the hyperplane separation theorem to show that the CPP algorithm will yield an expected core with high probability strengthens results.
- Connections drawn between the geometric properties of the game and its learnability form an important basis for theoretical analysis.

## Weaknesses
- Limited work on relaxations of the strict convexity requirement limit the scope of the paper.
- It would be important to discuss the use of these techniques in the motivating application domains mentioned in the paper. In general, there is a lack of empirical investigation and it is unclear how well the algorithms proposed would do in terms of sample efficiency, complexity, or robustness when key assumptions are violated.